# Low-Cost Microfabrication Tool Box

**DOI:** 10.3390/mi11020135

**Published:** 2020-01-25

**Authors:** Jérôme Charmet, Rui Rodrigues, Ender Yildirim, Pavan Kumar Challa, Benjamin Roberts, Robert Dallmann, Yudan Whulanza

**Affiliations:** 1Warwick Manufacturing Group (WMG), University of Warwick, Coventry CV4 7AL, UK; r.rodrigues@warwick.ac.uk; 2Mechanical Engineering Department, Middle East Technical University, 06800 Ankara, Turkey; yender@metu.edu.tr; 3Department of Chemistry, University of Cambridge, Cambridge CB2 1EW, UK; pkc31@cam.ac.uk; 4Warwick Medical School, University of Warwick, Coventry CV4 7AL, UK; b.roberts.2@warwick.ac.uk (B.R.); r.dallmann@warwick.ac.uk (R.D.); 5MRC Doctoral Training Programme in Interdisciplinary Biomedical Research, University of Warwick, Coventry CV4 7AL, UK; 6Department of Mechanical Engineering, Universitas Indonesia, Depok 16424, Indonesia

**Keywords:** microfabrication, microsystem, manufacturing, low-cost, scaling laws, lab-on-chip

## Abstract

Microsystems are key enabling technologies, with applications found in almost every industrial field, including in vitro diagnostic, energy harvesting, automotive, telecommunication, drug screening, etc. Microsystems, such as microsensors and actuators, are typically made up of components below 1000 microns in size that can be manufactured at low unit cost through mass-production. Yet, their development for commercial or educational purposes has typically been limited to specialized laboratories in upper-income countries due to the initial investment costs associated with the microfabrication equipment and processes. However, recent technological advances have enabled the development of low-cost microfabrication tools. In this paper, we describe a range of low-cost approaches and equipment (below £1000), developed or adapted and implemented in our laboratories. We describe processes including photolithography, micromilling, 3D printing, xurography and screen-printing used for the microfabrication of structural and functional materials. The processes that can be used to shape a range of materials with sub-millimetre feature sizes are demonstrated here in the context of lab-on-chips, but they can be adapted for other applications. We anticipate that this paper, which will enable researchers to build a low-cost microfabrication toolbox in a wide range of settings, will spark a new interest in microsystems.

## 1. Introduction

Microsystems are miniature devices typically made up of components between 1 and 1000 µm in size. They usually consist of moving or static mechanical and electrical parts that can interact with their surroundings. These features, combined with the possibility to integrate them with modern semiconductor technologies, makes microsystems excellent microsensor or microactuator candidates. Due to their small size, they can be mass manufactured at low unit cost via parallel processing techniques and they can be integrated seamlessly with other devices. Examples of microsystems include accelerometers, gyroscopes, pressure sensors, micropumps and gravimetric sensors, which are now found in a number of consumer products, including mobile phones, cars, or energy harvesting and medical devices. The manufacturing of microsystem evolved from the semiconductor fabrication processes initially developed to fabricate integrated circuits [1]. While the early microsystems were made of silicon or other semiconductors, polymers, metals and ceramics are now routinely used [1].

The fact that such microsystems are manufactured and operated at the microscale challenges our approach to fabrication technologies and our conception of the influence of physical forces at play. Before discussing the fabrication technologies, and in particular low-cost technologies, which is the focus of this paper, we give a brief overview of the scale dependence of physical forces and their effect on microsystems. In particular, we focus on effects that dominate at the macroscale and become negligible at the microscale and vice versa. Such effects can have positive or negative impacts on the operation of microsystems as illustrated in three example:Heat exchange. Due to the large surface to volume ratio, the heating or cooling of microdevices happens on a much faster timescale. This property is useful for the development of micro-heating elements with fast response and uniform temperature distribution, for example, in the context of gas sensing [2].Surface forces. The large surface to volume ratio of microdevices implies that surface forces play an important role at the microscale. One consequence of this is the difficulty to use conventional mechanical motors and coupling, such as gears, to induce motion due to the large forces needed to overcome stiction. The most common approaches to induce motion in microsystems rely on piezoelectric, capacitive, or magnetic actuation mechanisms [1].Laminar flow. In the case of fluid, flow regimes are typically laminar for aqueous solutions at the microscale. This is due to the fact that, at this scale, viscous forces dominate over inertial forces. The Reynolds number that represents this ratio is used to specify the transition between turbulent and laminar flow. It is defined as Re = *uL*/*ν*, where *u* is the flow velocity of the single-phase fluid; *L*, the characteristic length scale and *ν*, the kinematic viscosity. In the laminar regime, occurring at low Reynolds numbers, the flow properties can be quantified easily and manipulated in a controlled way [3]. This feature is used in microfluidics, for example, for diagnosis and samples processing applications [4].

The three examples above can be formalised using the scaling laws, which describe the relationship between two physical quantities that scale with dimensions, and can be used to understand the transition from macroscopic to microscopic dimensions [1,5].

Having highlighted some of the advantages of microsystems, let us now evaluate their fabrication. As mentioned above, the processes used to manufacture such devices have evolved from semiconductor industry processes. Due to the size constraints and the necessity to operate in a contamination-free environment, the fabrication of microsystems typically relies on highly specialised equipment usually operated in a clean room environment. Therefore, the cost associated with conventional microsystems fabrication is high and has restricted the development (and teaching) of microsystems to specialised laboratories and institutions in upper-income countries. However, recent technological advances and the recognition that some processes do not need to be operated in a clean room has paved the way for the development and implementation of low-cost microfabrication tools. Clean rooms are necessary to maintain high production yield, however, this constraint can be relaxed to some extent (as a rule of thumb, the environment should be as clean as possible) for research and development purposes.

In this paper, we report a range of low-cost microfabrication approaches and equipment developed or implemented in our laboratories. We detail the processes and provide all the information to build or adapt the tools for the microfabrication of structural and functional materials capable of reaching sub-millimetre feature sizes (see Figure 1). In particular, we have selected, developed or modified equipment to stay below £1000 per tool, set as an arbitrary cost limit. For each technique and equipment presented, we highlight the advantages and limitations and we provide examples of similar approaches elsewhere. In particular, we report a low-cost ultraviolet light-emitting diode (UV LED) lithography set-up for single layer exposure; a “per object” process to optimize high resolution fused filament 3D printing; a low feed rate process as a method to compensate for the low rotation speed for low-cost micromilling, a process to cut thick polydimethylsiloxane (PDMS) sheets using xurography; and a low-cost screen printing rig based on a conventional table-top computer numerical controlled (CNC) router.

Due to the research focus of our labs, many examples are borrowed from the fabrication of microfluidic or lab-on-a-chip devices, but the techniques described in the manuscript can be used to fabricate other microdevices. 

## 2. Materials and Methods

Table 1 summarizes the list of equipment described in the manuscript, including indicative prices and suppliers. More information, including the bill of materials (BOM), circuit diagram, and specific process parameters are found in Appendix B, Appendix C, Appendix D, Appendix E and Appendix F as listed in Table 1. Further information, including code, settings and videos are provided in Appendix A.

## 3. Results and Discussion

In this section, we describe low-cost equipment and associated processes for the sub-millimetre patterning of materials. We cover both structural materials (i.e., materials that are typically used to provide a mechanical structure to support other devices or mask certain area of a given substrate) and functional materials (i.e., materials that confer function to a device, including, e.g., organic or metallic inks used as electrodes for heating, electronic connection or sensing).

### 3.1. Photolithography

Photolithography, which relies on light-to-pattern photosensitive materials, is one of the central processes used in the fabrication of integrated circuit (IC) and microsystems [6,7,8,9]. The photosensitive material, typically a photoresist, is UV irradiated through a photomask and developed to form three-dimensional structures on the substrate. In the case of negative photoresist, the UV irradiation initiates polymerisation, thus preventing the dissolution of the exposed areas when soaked into the development solvent [1].

Usually, the radiation in the UV range is achieved using gas-discharge lamps. Such sources produce a wide spectrum of light that needs to be filtered to the selected wavelength. Moreover, these lamps have a limited lifetime of approximately 2000 h, and require regular calibration, significant time to warm up and cooling during operation. Conventional mask aligners that allow for UV irradiation and positioning of photomask to enable multilayer patterning typically cost in excess of £ 50 k.

Building up on a low cost UV light-emitting diode (LED) wafer scale mask aligner that enables sub-micron patterning with alignment resolution lower than 10 µm that we developed previously [10], we propose a low-cost version set-up with an Arduino based timer for single layer exposure.

At the core of the set-up (Figure 2) is a 365 nm UV LED that has a 20,000 h lifetime, does not need warming time or cooling and whose power output is constant, removing the need for calibration. Combined with a collimating lens, we have used the lamp to successfully fabricate single layer master moulds using SU8 2000 and 3000 series photoresists (Microchem Corp., Westborough, MA, USA) and acetate or chrome masks (Microlitho Ltd., Chelmsford, UK). Low-cost photomasks can also be printed using inkjet printing on acetate [11]. The moulds were used to fabricate PDMS microfluidics devices (see e.g., [12,13]) using a replication moulding approach also known as soft-lithography [8]. It is noted that the same set-up can be used with other photoresists, for lift-off [1] to deposit electrodes or other functional materials for example.

The set-up comprising UV LED, optics and an Arduino microcontroller (see Appendix B) costs less than £ 800 and enables to reach feature sizes comparable to conventional UV photolithography [1]. In particular, it allows for the development of features down to 5 µm with acetate masks and is expected to reach sub-micron sizes with a chrome mask in contact mode and appropriate photoresist. To obtain high reproduction fidelity of the features on the mask, it is important to minimize the gap between the mask and the wafer. This can be achieved for acetate masks by stacking a 5 mm thick quartz window, the acetate mask (with printed features against the photoresist), and the coated wafer (in this order) onto a thin sponge covering the entire surface of the wafer. For higher resolution, using e.g., a chrome mask, the mask should be pressed more actively against the wafer using additional weight or a clamp. The addition of a xyz-stage and appropriate mask holder would enable multiple mask processes.

### 3.2. Three-Dimensional (3D) Printing

Three-dimensional (3D) printing is a process for building 3 dimensional objects from computer aided design models. Consumer grade 3D printing platforms suit the fabrication of devices with features larger than 100 µm in the XY plane and as low as 35 µm in the Z axis. Custom built equipment can perform well at even higher resolution [14] but will typically require extensive knowledge of the various components of a 3D printer (software/mechanics/fluidics) and are less likely to result in broadly reproducible designs.

Several 3D printing technologies may be used to achieve sub-millimeter features, but low cost (<£1000) applications limit the market to fused filament fabrication (FFF) and liquid crystal display (LCD) based stereolitography (SLA) printers. FFF relies on the deposition of fused polymer filament in stacks to produce 3D shapes, while SLA-based printing relies on stacks of photosensitive resin polymerized by exposure to a defined wavelength of light.

Low-cost LCD-SLA printers can outperform FFF printers in feature size but require fine tuning of the printing process depending on the resin used and layer height. FFF has an overall lower cost of equipment and consumables, lower complexity for setup and is supported by a larger community base. Waste produced by SLA printers requires careful handling due to its toxicity, and post-processing for removal of uncured resin typically requires the use of Isopropanol. This results in added requirements to meet health and safety and waste disposal regulations. These factors have led us to focus on the development of methodologies based on FFF, and this will be the focus of this section.

The last decade has seen explosive growth of low-cost FFF-based 3D printers. Competition has driven hardware suppliers to diversify their offer, integrate new sensors and develop improved control software. These improvements have focused on broadening material selection and integration of multiple materials, perfecting the level of detail of printed parts, increasing reliability and extensive correction of known issues leading to suboptimal surface finish. The open nature of the community has also led to the cross-integration of improvements and extensive networks of suppliers. FFF can be used to quickly prototype microfluidics systems either through direct production of devices or creation of moulds for polymerisation of PDMS [15,16].

However, FFF printers suffer many drawbacks for microfluidic device fabrication, including surface roughness, and filament stock variability (requiring calibration when filament supplier/material/lot changes). Moreover, the control of filament extrusion remains a challenge using non-specialist software and hardware as variations of internal nozzle pressure are amplified when using small diameter nozzles [17]. One way to reduce these variations is to slow down the extrusion through overall decreased printing speed and reduced acceleration per axis. However, this approach leads to extended printing times which is not practical for large prints.

We have addressed these issues by proposing the introduction of “per object” settings to bring more flexibility to the mould-printing process. Here, we purpose an approach in which three separate objects (base, device and wall) are used to allow for iteration through print settings. Using this approach, a base for the device can be produced quickly with a desired thickness, and top layer smoothing is done through a process called ironing. This process involves a second pass of the hot nozzle over the printed area with a much reduced filament flow to allow the nozzle margins to remove excess material left during the initial pass, as well as maintain minimal flow to ensure a smooth finish. This smoother surface enables better contact of PDMS devices to other surfaces avoiding leaks. Walls can be produced with standard extrusion settings. Finally, the device itself can have a dedicated set of settings to optimize dimensional accuracy. In complex devices these layers can be further broken down into separate objects in order to vary settings as required.

The process parameters reported here have been optimized for the MK3 variant of the Prusa i3 (Prusa Research, Prague, Czech Republic) and a 0.2 mm nozzle diameter that enabled us to consistently fabricate moulds with features of approximately 200 µm (Figure 3). Using the Cura slicer (Ultimaker BV, Utrecht, The Netherlands), and RS Pro 1.75 mm acrylonitrile butadiene styrene (ABS) filament (RS Components, Corby, UK) moulds are printed and PDMS-based devices replicated using soft-lithography [8]. SEM images of moulds and devices can be seen in Figure 3a–f. It is noted that the shape of extruded sections is also substantially different from typical rectangular channels obtained through photolithography, and this factor should be taken into account when modelling microfluidic devices. Appendix C includes more details about the process and the profiles used for printing of the mould shown in Figure 3.

### 3.3. Micromilling

Micromilling is a vertical milling process for machining micro-cavities such as microchannels and microreservoirs by using end mill cutters with diameters of less than 1 mm (typically in the order of 100 µm). A prototype can be fabricated in less than 1 h after the design stage [18]. Micromilling is widely used for direct machining of microfluidic devices on polymer substrates. In this context, it was shown that the minimum feature size is the same as the tool diameter as demonstrated with polystyrene (PS), polymethylmethacrylate (PMMA), and cyclic olefin copolymer (COC) by using 127 µm diameter end-mill [18]. Moreover, micromilling of PMMA by using end-mills with diameter as low as 20 µm have also been demonstrated [19]. On the other hand, the process can also be utilized to fabricate moulds for embossing [20] or PDMS moulding.

The principles of operation of micromilling are essentially the same as for the conventional vertical milling process. However, the dimensions involved in micromilling (small diameter of the cutter, low uncut chip thickness depending on the feed rate) strongly affect the characteristics of the process. For example, cutting forces increase with increasing chip thickness (*t_c_*) [21], as given by *t_c_* = *f_t_*/*ω*, with *f_t_* being the feed per tooth on the cutter (the feed divided by the number of cutting teeth) and *ω* being the rotational speed of the cutting tool. Therefore, to reduce the risk of fracture failure of the cutting tool, rotational speed of the cutting tool is typically increased to maintain low lead time. However, high-speed (up to 100,000 rpm) milling machines typically cost more than ~£10k.

Since the lead time is not of major importance for research and development purposes, we propose to decrease the feed rate instead. In particular, we have shown that a desktop milling machine with an acceptable positioning accuracy (depending on the application) can be utilized for micromilling. We have fabricated micromixers by feeding 200 µm diameter end-mill, rotating at 2000 rpm, at 5 mm/min against PMMA blocks using a desktop milling machine (ProLight Machining Center WPLM1000, Light Machines Corp., Manchester, NH, US) [22]. Figure 4 depicts a low-cost (~£ 500) milling machine (Proxxon MF70 CNC-ready, Wecker, Luxemburg) with 20,000 rpm spindle and 5 µm resolution machining a microfluidic device. An example, with corresponding part program, and process parameters is presented in Appendix D.

Although low-feed machining, characterized by feed rates less than or equal to 10 mm/min, solves the need for high-cost equipment with spindle speeds greater than 20,000 rpm, it may also introduce new challenges such as overheating of the cutting tool and the substrate, which may lead to increased burr formation. We have tested milling of 400 µm channels at spindle speeds ranging between 5000 rpm and 20,000 rpm at a fixed feed rate of 3 mm/min. We observe on Figure 4b that a spindle speed of 20,000 rpm produces burr-free channels, however, we note that the burr formation increases as the spindle speed is reduced as hypothesized.

### 3.4. Xurography

Laser machining is widely used to prototype microdevices, including open-channel microfluidic devices [23]. However, lasers are expensive and require skilled operators. Desktop cutters, such as the Silhouette Curio™ (Figure 5a) that costs ~£200 and is easy to setup up and run, appears as an interesting low-cost alternative. Open structures such as channels, can be easily designed on the free software Silhouette Studio^®^, which allows for design features down to 100 μm in thin films [24]. The incisions are made with a sintered tungsten alloy blade (Figure 5b,c) into the desired material.

Xurography has been used to cut cyclo-olefin-based flow chambers to study microtubules dynamics under mechanical stress [25], cut omniphobic fluoroalkylated paper (R^F^ paper) in the production of microfluidic paper-based analytical devices [26] and create moulds for soft-lithography [27]. Even though xurography can achieve good fidelity and reproducibility when cutting relatively thin (<200 μm) and rigid sheets of materials, the use of thick and soft materials (such as PDMS) is more challenging as the sheets deform during the cut process [24].

Here we report a novel approach that consists in adding Kapton^®^ (polyimide) adhesive tape to cover the PDMS sheet prior to cutting. This approach provides more rigidity to the material, and reduces the risk of movement during the cut process, producing more uniform channel width and wall structure with cut accuracy comparable to that obtained using rigid sheets. We noted a 17% (+/- 9%) variation in channel width for the PDMS sheet alone (5 samples) and 7% (+/- 4%) when adding Kapton^®^ on top of the PDMS sheet (5 samples) compared to the same channel cut in Kapton^®^ tape (average of 10 samples, 500 μm channel width). The addition of the tape also acts as a structural support during the transportation of the PDMS post cut.

We have cut sheets of 500 μm thick PDMS (1:10 ratio of curing agent to base from Sylgard 184) with open channels. Such structures are typically used for the production of organ-on-a-chip devices requiring overlapping channels. With the blade adjusted to a height of 500 μm, speed and force of cut set to 2 and 20 respectively (arbitrary from Silhouette Studio^®^), the program was run and the incisions were made on PDMS covered with tape, creating open channels of 500 μm depth and 500 μm width (Figure 5e).

### 3.5. Screen Printing

Screen printing is a technique used to deposit ink through structured screen meshes. A blade or squeegee is moved across the mesh to fill its open apertures with ink. The pressure on the blade puts the mesh into contact with the substrate, which results in the transfer of ink patterns as shown in Figure 6. One of the main application of screen printing is the deposition of electrodes for electrochemical sensing [28]. Recently the technique has been used for wearable sensors [29] and photovoltaic [30] applications as well as for flow sensing where inks are used as heating and sensing elements [31].

Screen printing is typically used to produce patterns with lateral feature sizes below 100 µm and thickness between 5 and 100 µm [32]. The structured screen mesh is usually fabricated using photosensitive emulsion scooped across the mesh, and exposed through a photomask using a photolithography approach. In our case, the UV LED set-up described above has been used to fabricate the screen meshes from T120 polyester screen from Nectex with wire mesh of 45 ± 1 µm. Of note, the resolution of the patterns is determined by the wire spacing of the meshes used.

Our screen printing rig was assembled for less than £ 300 by modifying a standard 3-axis-mini-CNC router. The modifications consisted of customized parts to accommodate a squeegee on the x-axis and a frame to attach the screen to the chassis of the CNC router (Figure 6a). The distance between the squeegee and the silk screen was set between 0.15 and 0.4 mm. The displacement and speeds are adjusted to provide enough pressure to allow for screen printing on the printed circuit board (PCB) after loading the ink on the screen. Figure 6b and c show an example of carbon-based electrodes deposited using this rig, the assembly of which is detailed in Appendix F.

## 4. Conclusions

In this paper, we have presented a range of low-cost microfabrication approaches and equipment, developed or implemented in our laboratories. In particular, we have covered photolithography, micromilling, 3D printing, xurography and screen-printing. Some platforms were fabricated and assembled by us (e.g., photolithography set-up), others were bought off-the-shelf (e.g., desktop cutter) and some resulted from mechanical modification or parameter adaptation of existing equipment (screen printing, 3D printing and micromilling device). Importantly, all of them address the need for low-cost microfabrication tools capable of generating patterns with sub-millimeter feature sizes.

The techniques and devices presented can be combined to fabricate complex devices, such as lab-on-a-chip platforms. For example, an amperometric-based platform can be fabricated by combining a microfluidic channel network (fabricated using replication moulding of PDMS over a mould realized using micromilling, 3D printing or photolithography) and three electrodes for electrochemical sensing (deposited via screen printing on a PCB). The PDMS microfluidic chip can then be secured over the PCB using a 3D printed clamp. In addition, by providing all information necessary to build and/or operate the various pieces of equipment described in the paper, following the open source model, we anticipate that some readers will modify and adapt them to suit further specific requirements.

Microsystems are key enabling technologies, and are widely used in in vitro diagnostic (IVD), energy harvesting, automotive and navigation, telecommunication, drug screening, etc. With this paper, we aim to lower the entry barrier (for both research and educational purposes) into the fascinating and highly promising world of microsystems.

## Figures and Tables

**Figure 1 micromachines-11-00135-f001:**
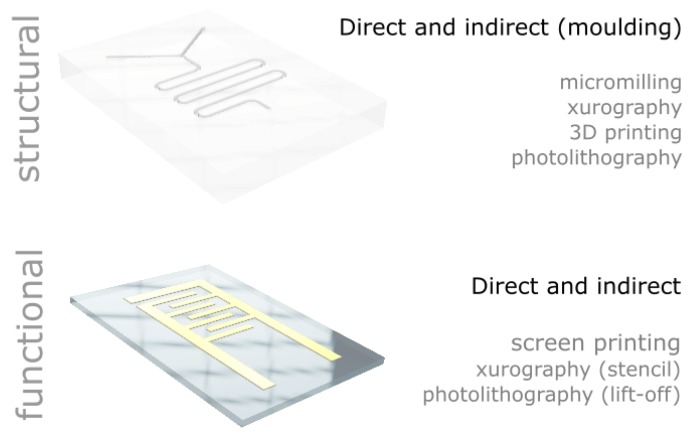
Overview of the low-cost processes presented in the manuscript for the structuring of structural and functional material with feature sizes below 1 mm.

**Figure 2 micromachines-11-00135-f002:**
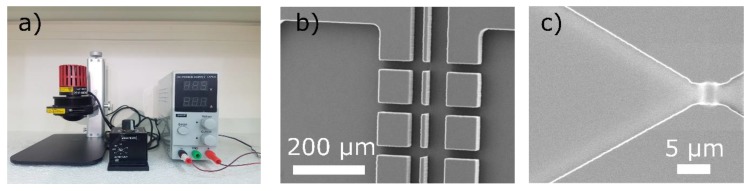
Low-cost photolithography set-up. (**a**) Ultraviolet light-emitting diode (UV LED) set-up at Universitas Indonesia, for single layer exposure. (**b**) Scanning electron microscope (SEM) micrograph of a polydimethylsiloxane (PDMS) microstructure replicated from a master mould (SU8 3025 on silicon wafer). (**c**) SEM micrograph of SU8 (2000.5) structures obtained using a chrome mask.

**Figure 3 micromachines-11-00135-f003:**
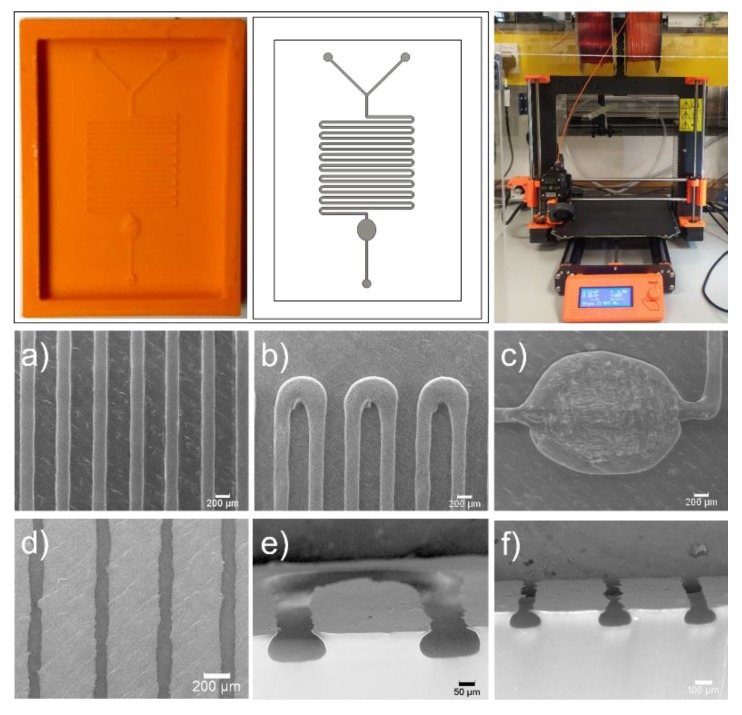
Example device in acrylonitrile butadiene styrene (ABS), with corresponding design and photo of the Prusa i3 MK3 equipped with 0.2 mm nozzle (**top**) and SEM images of mould (panels a, b and c) and PDMS replica (panels d, e and f) features. (**a**) channel sizing and parallelism; (**b**) effect of curved path on filament deposition; (**c**) effect of multiple concentric lines on round features; (**d**) visibility of ironing path on replicated devices; (**e**) elliptical nature of filament deposition and surface smoothness; (**f**) channel-to-channel variation.

**Figure 4 micromachines-11-00135-f004:**
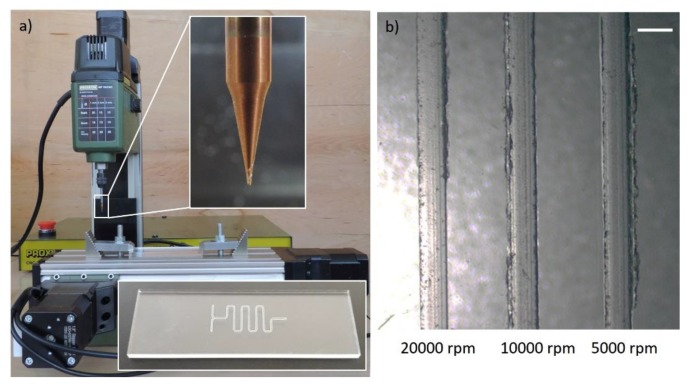
Proxxon MF70/CNC-Ready micro milling machine at Middle East Technical University, Mechanical Engineering Department and close-up views of 400 µm diameter end-mill and a sample microfluidic device (serpentine micromixer) milled on PMMA substrate. (**b**) Microscope view of channels micromilled at 20,000 rpm, 10,000 rpm and 5000 rpm spindle speeds after 2 min sonication. Feed rate was fixed at 3 mm/min. Increasing burr formation with decreasing spindle speed implies overheating problems at low-feed (less than 10 mm/min) milling at low spindle speeds. Scale bar shows 500 µm.

**Figure 5 micromachines-11-00135-f005:**
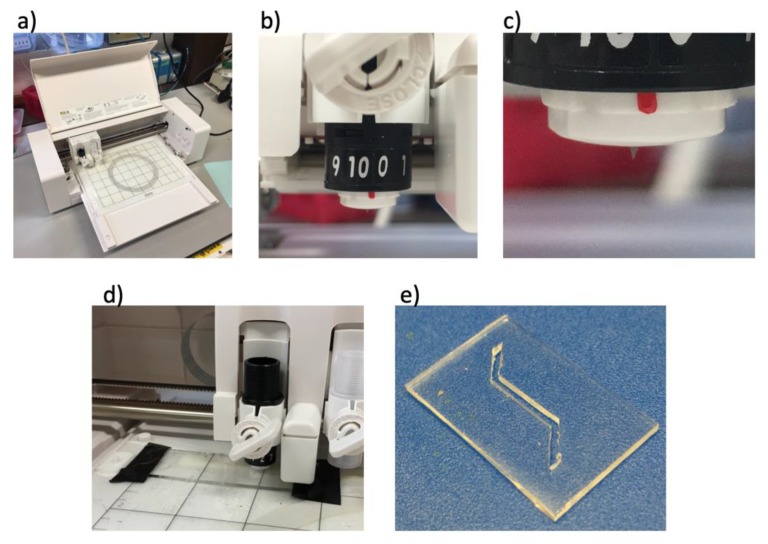
Low cost desktop cutter. (**a**) The desktop Silhouette Curio™: (**b**) Tunable tungsten alloy blade used to make incisions. (**c**) Sintered tungsten alloy blade set to 1000 μm cut depth. (**d**) Silhouette Curio™ cutting into a 500 μm sheet of PDMS. (**e**) A PDMS microfluidic channel successfully cut using the Silhouette Curio™. Height and width of the channel were set to 500 μm.

**Figure 6 micromachines-11-00135-f006:**
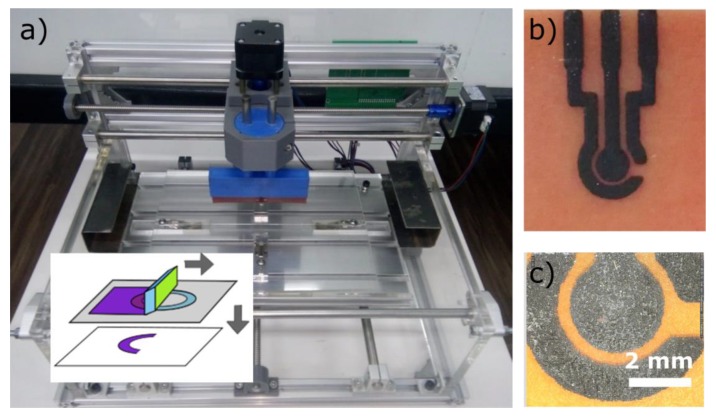
Screen printing. (**a**) Screen printing device and principle of operation (inset). (**b**,**c**) carbon-based electrode deposited using screen printing at two magnifications.

**Table 1 micromachines-11-00135-t001:** List of equipment mentioned in this paper.

*Equipment*	Ultraviolet Light Emitting Diode (UV LED) Lithography	3D Printer Fused Filament Fabrication (FFF)	Milling Machine	Cutting Plotter (Xurography)	Screen Printing
Cost	<£ 800	~£ 600	~£ 400	~£ 200	<£ 300
Evaluated feature size (this manuscript)	5 µm (with acetate mask)	220 µm with 0.2 mm nozzle diameter.	400 µm	500 µm	500 µm
Minimum feature size	<1 µm (with chrome mask)	~100 µm with 0.1 mm nozzle *	100 µm	100 µm	30 µm
Resolution limitations	Function of mask and photoresist	Function of nozzle diameter and feature size	Function of tooling diametre	Function of the rigidity and thickness of the film	Function of screen mesh size
Model and instructions	Custom built Bill of materials (BOM), circuit diagram and code in Appendix B	Prusa i3 MK3 Example process parameters in Appendix C	Proxxon MF70 CNC-ready Example process parameters in Appendix D	Silhouette Curio Example process parameters in Appendix E	Custom built rig Assembly instruction in Appendix F
Typical materials	UV sensitive resin (e.g., SU8, AZ® series)	Acrylonitrile butadiene styrene (ABS), Polyethylene Terephthalate-Glycol (PETG)	Polycarbonate (PC), Polystyrene (PS), Polymethyl methacrylate (PMMA), Cyclic olefin copolymer (COC)	Acetate film, polyimide adhesive film, PDMS sheet	Silver or carbon ink

***** For positive features, scales with nozzle diameter/negative features depend on printer resolution per axis.

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
