# Peer review of "Low-Cost Microfabrication Tool Box"

_micromachines, 2020, doi:10.3390/mi11020135_

Round 1

Reviewer 1 Report

The article by Charmet et al. reviews 5 different techniques in use in their labs for low-cost microfabrication. I think this article will be useful for other researchers and recommend it for publication -- maybe more as a "mini-review" than as a "research article", although this is an editorial choice.

I have only minor comments.
- it would be useful to describe in the appendix the assembly of the setup for UV photolithography, and not give only the BOM. Although this setup may already be described in a previous article, a summary of the main points would be nice.
- concerning 3d printing: PDMS is usually cured at 70 degrees C to polymerise in a couple hours. Have the authors ever noticed that 3d-printed moulds deform at this temperature? This would be something to mention in the text, or if lower temperatures are used to cure the PDMS, they should also mention it.
- can the authors give the characteristics of the screen used for screen printing?

Author Response

We thank the reviewer for their positive feedback. We have addressed the comments below and updated the manuscript accordingly.

We have added a diagram for the UV photolithography set-up in Fig A2. It shows how the various components (from BOM) are assembled and the connection to the Arduino set-up. We have also updated the description of Appendix A accordingly (see lines 343-344 in the manuscript).

We thank the reviewer for pointing out missing information about the PDMS curing process. We are indeed avoiding the deformation of the mould by fixing it to a flat substrate (typically a glass slide) using epoxy resin. We have now updated the list of materials used and added the following information to Appendix B (lines 378-385, highlighted in red).

Note for PDMS curing process:

              In order to avoid mould deformation under typical PDMS curing conditions (60-70°C) the bottom surface of the mould should be fixed onto a flat surface using an appropriate epoxy resin.

Also, we do not recommend the use of PLA to fabricate moulds due to its low glass transition temperature, and therefore we have mentioned it from the updated Table 1 that now mentions target materials.  

Regarding the screen used for the screen printing rig, we have used a T120 polyester screen with wire spacing of 45±1µm, which corresponds to the printing resolution. We also note that the structuring of the screen was done using the UV lithography set-up reported in the manuscript. We have updated the following sentence in the manuscript (line 293-294).

In our case, the UV LED set-up described above has been used to fabricate the screen meshes from T120 polyester screen from Nectex with wire mesh of 45±1µm.

Reviewer 2 Report

The manuscript by Charmet et al. is presenting a recapitulation of previous experience by the same authors on developing alternative low-cost solutions for micro-fabrication processes. The topic is of potential interest for a wide audience: state-of-the-art micro/nano-fabrication facilities are often too expensive and out of reach for many scientists who would still intend to develop micro-systems for their research. Therefore, a review of such alternatives is potentially very useful and I strongly appreciate the intention of the authors.

However, I don't find the manuscript suitable for publication in its present form. The content is almost completely a review of already published results from the same authors but it is proposed as an article. it should instead be framed as a formal review and a discussion of other relevant examples from literature (and different groups) needs to be added for completeness. The introduction also suffers from lack of depth in discussing scaling-laws and how they affect the principle of design for micro-systems: if such introduction to the merits of micro-system needs to be kept, then it must be developed to a deeper extent. It is equally acceptable, in my opinion, to completely remove this part instead. Finally, in view of expanding this contribution to a suitable review, re-organizing the material in terms of applications rather than techniques would be preferable, as for example what are the standard technique to fabricate microfluidic devices and the benefit/limitations of alternative cheaper approaches.

Author Response

We thank the reviewer for their feedback and the opportunity to improve our manuscript. We have answered all the questions/issues raised by the reviewer in the manuscript and below.

Our reasons for not proposing a review are highlighted below:

we are presenting novel processes / methods from our laboratories. However, from the reviewer’s comment, we realise we have not managed to convey the novelty of our methods/process in the manuscript. Therefore we have updated the manuscript to make the innovations clearer and we provide a summary below for the five techniques: Even though we have published about a wafer-scale UV LED lithography system, the one we propose in this manuscript is a stripped down version that costs less than £1000 and enables single layer exposure only. Compared to the previous set-up [ref 6], we redesigned the rig, wrote a new code for an Arduino based timer –instead of a Raspberry Pi and presented new designs. We have updated the manuscript (lines 146-149, highlighted in red). For the 3D printing, we are presenting a novel approach, consisting in separating the mould into 3 separate object to enable the optimisation of separate process parametres for each object. This approach has enabled us to obtain a smooth finish (for the based) and small, yet consistent feature sizes (for the channel). We have updated the manuscript (lines 168-183). We are reporting a low feed rate as a method to compensate for the low rotation speed in micromilling. Even though we have used a low rotation speed for the fabrication of a micromixer previously [ref 20 in manuscript], it was due to lack of access to a high speed micromilling machine. After the paper was published, we have investigated the process in more details and confirmed its “universality”, hence our interest to present our findings here to highlight this low-cost option. We have updated the manuscript (lines 248-254). Even though xurography has been used to create microfluidic device before, it is the first time, to the best of our knowledge, it is used to cut thick PDMS sheets (it is normally used to cut thinner rigid polymer sheets). We report here an improved process using an adhesive tape that enables the cutting of thick PDMS layers. We have updated the manuscript (lines 264-275). Finally, even though the screen printing device is inspired by similar device, we have optimised it for low cost and in particular the screens are prepared using the UV LED photolithography set-up presented in the manuscript. We have updated the manuscript (lines 292-293). It would not be appropriate (or even possible) to provide BOMs, codes, parameters, etc. for processes or devices from other laboratories (in case we had decided to propose a review). Therefore it would defeat the purpose of the manuscript, which we hope can trigger new interest in microsystems (and microfluidics), by providing people with low-cost microfabrication tools.

We have therefore updated the paper to emphasize the novelty for each of the equipment/method as mentioned above, and we have also added a short section in the introduction (lines 87-97) and shown below:

In this paper, we report a range of low-cost microfabrication approaches and equipment developed or implemented in our laboratories. We detail the processes and provide all the information to build or adapt the tools for the microfabrication of structural and functional materials capable of reaching sub-millimetre feature sizes (see Fig. 1). In particular, we have selected, developed or modified equipment to stay below £1000 per tool, set as an arbitrary cost limit. For each technique and equipment presented, we highlight the advantages and limitations and we provide examples of similar approaches elsewhere. In particular, we report; a low-cost UV LED lithography set-up for single layer exposure; a “per object” process to optimize high resolution fused filament 3D printing; a low feed rate process as a method to compensate for the low rotation speed for low-cost micromilling, a process to cut thick PDMS sheets using xurography; and a low-cost screen printing rig based on a conventional table top CNC router.”

Regarding the information about the scaling laws, we feel it is important to mention them in the manuscript. Again, we would like to reiterate the fact that the intention of the manuscript is to trigger new interest in microsystems, therefore it is also aimed at people who may not be aware of scaling laws. As such, we feel that a short introduction (and links to more in-depth analysis see e.g. [ref 5]) is appropriate. We have also added a link to [ref 1] that also describes scaling laws.

Regarding the re-organisation of the material, we feel that it is more appropriate to keep it the way it is. We feel that the toolbox approach (in particular for a novice who may not be familiar with  microfabrication technologies) opens up more opportunities as the tools will not be linked to a single application. However, the comment of the reviewer is pertinent and our structure could indeed confuse more experienced readers. Therefore, we propose to add another figure (now Figure 1) that shows how the various techniques can be used to create structural and functional microstructures. This will help to reinforce the toolbox approach and avoid confusion. We note also that we have updated all figures numbers accordingly.

Reviewer 3 Report

This paper falls somewhere between a research paper and a review, but unfortunately it does not accomplish its goals as either one. The paper uses good English and targets an interesting area, which is clearly in the scope of this journal. The appendices are a nice touch, giving readers the “recipes” to create their own systems. But in the end, it reads more like an online tech blog, rather than a scientific paper. For the latter, it would require fundamental revisions (a resubmission after rejection). The major drawback is the lack of quantitative analysis on the techniques, which prevents a comparison between the techniques presented. Therefore, the work seems like a few unconnected “stories” and not a scientific paper.

Data must be provided: I cannot confirm from this paper, for example, that the (controversial) claim of sub-micron resolution can be achieved by the photolithography system described. The same problem is observed throughout.

Qualitative statements should be avoided. I give one example, but similar ones can be found throughout: "Although low-feed machining solves the need for high-cost equipment, it may also introduce new challenges such as overheating of the cutting tool" (specific range of feed rates required)

Better/rigorous comparison between techniques is required.

Table 1. Price is fine, but the “capabilities” are not clear. Eg. What do the authors mean by “resolution” and “minimum feature size”? These terms need to be rigorously defined and data to support their values need to be provided. It seems to me that these values were obtained from the manufacturer or perhaps another source?

Add two tables:

1-list the relevant target materials by micromachining method.

2-compare figures of merit between the approaches considered in this paper. Examples could include "fidelity" between the design (i.e. channel width as a percent of the design size), wall roughness, etc.

In summary, I must recommend this paper for rejection.

Author Response

We thank the reviewer for their feedback and highlighting shortcomings in our manuscript. We would also like to thank them for giving us the opportunity to improve it. As per the editor’s guidance, who requested a resubmission within 10 days, we have answered all the questions/issues raised by the reviewer below and updated the manuscript accordingly. In particular, we thank the reviewer for requesting further clarification about the minimum feature size and the addition of information in tables, which -we agree- will strengthen the manuscript.

First, the reviewer mentions that the manuscript reads like a blog, rather than a scientific paper. We believe that this claim is related to the main issue highlighted by reviewer 2 and is because we have not managed to convey the novelty of our equipment/process appropriately. Therefore, we have updated the manuscript to make the innovations clearer and we provide a summary below for the five techniques.

Even though we have published about a wafer-scale UV LED lithography system, the one we propose in this manuscript is a stripped down version that costs less than £1000 and enables single layer exposure only. Compared to the previous set-up [ref 6], we redesigned the rig, wrote a new code for an Arduino based timer –instead of a Raspberry Pi and presented new designs. We have updated the manuscript (lines 146-149, highlighted in red). For the 3D printing, we are presenting a novel approach, consisting in separating the mould into 3 separate object to enable the optimisation of separate process parametres for each object. This approach has enabled us to obtain a smooth finish (for the based) and small, yet consistent feature sizes (for the channel). We have updated the manuscript (lines 168-183). We are reporting a low feed rate as a method to compensate for the low rotation speed in micromilling. Even though we have used a low rotation speed for the fabrication of a micromixer previously [ref 20 in manuscript], it was due to lack of access to a high speed micromilling machine. After the paper was published, we have investigated the process in more details and confirmed its “universality”, hence our interest to present our findings here to highlight this low-cost option. We have updated the manuscript (lines 248-254). Even though xurography has been used to create microfluidic device before, it is the first time, to the best of our knowledge, it is used to cut thick PDMS sheets (it is normally used to cut thinner rigid polymer sheets). We report here an improved process using an adhesive tape that enables the cutting of thick PDMS layers. We have updated the manuscript (lines 264-275). Finally, even though the screen printing device is inspired by similar device, we have optimised it for low cost and in particular the screens are prepared using the UV LED photolithography set-up presented in the manuscript. We have updated the manuscript (lines 292-293).

We have also also added a short section in the introduction (lines 87-97) and shown below to emphasize the novel equipment/processes:

In this paper, we report a range of low-cost microfabrication approaches and equipment developed or implemented in our laboratories. We detail the processes and provide all the information to build or adapt the tools for the microfabrication of structural and functional materials capable of reaching sub-millimetre feature sizes (see Fig. 1). In particular, we have selected, developed or modified equipment to stay below £1000 per tool, set as an arbitrary cost limit. For each technique and equipment presented, we highlight the advantages and limitations and we provide examples of similar approaches elsewhere. In particular, we report; a low-cost UV LED lithography set-up for single layer exposure; a “per object” process to optimize high resolution fused filament 3D printing; a low feed rate process as a method to compensate for the low rotation speed for low-cost micromilling, a process to cut thick PDMS sheets using xurography; and a low-cost screen printing rig based on a conventional table top CNC router.”

The reviewer rightly noted the ill-defined defined “Capabilities column” in Table 1. They also noted the confusion arising between feature size and resolution and the fact that we have reported the “capabilities” without proving them. Finally, they also suggested to add information about the materials used and a more thorough comparison of figures of merit between the different techniques.

We have decide to address all of these issues by updating Table 1 as described below:

We have decided to include all the information (including target materials) in Table 1 (after switching from line to column format) to allow for better comparison of each approach proposed. We have removed the “capability” column and replaced it by 3 lines; namely “Evaluated feature size”, “Minimum feature size” and “Resolution limitations”. Doing so, we remove the ambiguity between feature size and resolution and we show clearly the feature sizes evaluated for the purpose of the manuscript. However, we feel it is important to highlight the capability of the techniques by reporting the minimum feature sizes that each tool is capable of reaching – based on existing literature (as reported in the manuscript – see below for details) so as not to restrict the scope of the manuscript. We also note that it is not our intention to compare the five techniques, that can either be used on their own (or in combination) to create a range of microstructures. Indeed one is subtractive (xurography) while the others are additive, all enable 2.5D (fixed height) structure except 3D printing, etc. Therefore, it would be difficult to compare other figures of merit. This is why we have chosen to only report on the feature size, which is common to all five techniques. Finally, we have reported the Resolution limitations to highlight the factors that will influence other figures of merit (e.g. the type of photoresist and its thickness will define the fidelity of the features compare to the mask). With the above, we hope that we have convinced the reviewer, that it would not be appropriate to attempts to compare other figures of merit, but we hope that the new table addresses their main concern.

Below we provide technique-specific information

UV lithography: we thank the reviewer for requesting a clarification about the feature size obtained by our UV lithography set-up. First, we would like to highlight the fact that resolution in photolithography is not limited by the light source (at a given wavelength) – the use of a UV LED is the only difference (in terms of light source) between our UV lithography system and conventional/ mask aligners. For positive resists, with chrome masks in contact mode, the minimum feature size is below 0.5 um [see e.g. ref1 in the manuscript]. The major limitation above this limit is the resolution of the masks. Low cost acetate mask typically have a resolution of ~5 um, which is what we have used for the structure shown in Fig. 2b. We note also that there are now a number of commercial mask aligners using UV LEDs. We have updated Table 1 and updated manuscript (lines 146-149 and below) to highlight these limitations:

The set-up comprising UV LED, optics and an Arduino microcontroller (see Appendix A) costs less than £800 and enables to reach feature sizes comparable to conventional UV photolithography [1]. In particular, it allows for the development of features down to 5 µm with acetate masks and is expected to reach sub-micron sizes with a chrome mask in contact mode and appropriate photoresist.

3D printing: would also like to highlight a few points regarding the surface roughness and the fidelity of the 3D printed moulds.  

Surface roughness in the bottom of moulds observed without the use of ironing as a print parameter will typically lead to the formation of unwanted microchannels. This prevents the use of 3D-printed moulds directly without extra post-processing steps. Ironing of the device layers will on the other hand typically result in deformed channels with irregular compression of materials at the top of the deposited filament.

Use of per object settings allows for differentiation of not only this parameter but also of those linked with material deposition (speed, acceleration/jerk, filament flow) and processing of the CAD model by software (gap filling, thin wall printing, maximum resolution, etc). Material deposition settings will contribute to the fidelity of reproduction of the CAD model during printing, while processing settings will ensure that the slicer will not add deposition or travel commands that would otherwise insert unwanted structures on the mould (again affecting fidelity).

Any of these strategies could be implemented in a custom slicing software package, but this would require extensive software development efforts and maintenance after software release, which is unlikely to occur unless adoption of these tools justifies it. We present an approach that enables researchers to produce functional moulds that will require minimal post-processing without the use of any proprietary platform. 

We have updated the section on 3D printing to convey the information more clearly (line 168-183).

Micromilling: we have now reported some more results about the influence of the feed rate (for a fixed spindle speed of 20000 rpm) on the quality of the cut. See updated Figure 4 and lines 248-256 in the manuscript. We have also added a few more references to highlight the capabilities of micromilling (lines 216-221).

Xurography: we have updated the text to highlight the novel process and also added some more information about the quality of the cuts (see line 264-275).

Screen printing, we have some information to highlight the fact that the screen was made using the UV lithography set-up reported in the manuscript and we have added information about the screen resolution (lines 292-293).

We hope that with the above, we have addressed the concerns of Reviewer 3.

Round 2

Reviewer 2 Report

The authors have addressed some of my concerns and provided an improved version of their manuscript. The intention on presenting technical solutions for low-cost micro-fabrication is more clearly emerging. However, for this manuscript to be accepted as a research paper I would expect the authors to address the following issues: 

UV lithography: resolution down to 5 um is claimed but not clearly shown, while below 1 um is proposed as possible but not otherwise substantiated. Claims as specific as these should not be presented without supporting evidence. In achieving high resolution when doing shadow printing-like UV-lithography, the assembly between mask and substrate is one of the most important process parameters (i.e. gap control) but here there is no mentioning of the mask/substrate assembling in this low-cost UV-lithography system. Please discuss if any clamping jig is used to help increase the resolution

3D-printing: it is obvious to users of 3D printing that FFF is the cheapest approach as compared to SLA, still this should be mentioned as the intention of this manuscript is to propose low-cost solutions. Limits and benefits of the two different system are worth a short discussion. The resolution capabilities of FFF 3D printing are stated as 100 um in XY, is this meant to be for positive or negative features? How does this limit varies with the nozzle size? Does the used material (PLA instead of ABS perhaps?) influence the resolution? Moreover, terms like "ironing" and "infill" are introduced but not defined in the main text, and infill seems misused in the caption of figure 3: infill' standard definition is the pattern and density of the structures inside the printed object, in the caption it is referred to as "multiple passes".

Xurography: this technique has been already proposed for cutting PDMS sheets in a similar way, see e.g. Lab Chip, 2009,9, 1290-129. The claimed improvement here refers to the thickness of PDMS layers successfully used.

Author Response

We thank the reviewer for their feedback and giving us an opportunity to improve the manuscript. We have addressed their concerns (see below) and updated the manuscript accordingly.

The reviewer requests further information about the resolution of our UV lithography set-up.

In particular, they rightly ask us to substantiate our claims for the fabrication of feature sizes down to 5 um. Our initial reluctance to show features sizes obtained using a chrome mask is that the addition of such a mask (>£400) brings us above the £1000 threshold we had set for each tool. However, after discussion with the co-authors, we have agreed on the fact that the “mask” is not part of the set-up, but it should rather be considered as a “consumable/supply”. As a consequence, we have now updated Figure 2 to show sub 5 um feature sizes using a Chrome mask and updated the manuscript accordingly.

The reviewer is also right to mention the importance of minimising the gap between the mask and the (coated) wafer. Even though we have omitted to mention it in the manuscript, it is indeed an important parametre. In our case, we place the coated wafer onto a piece of foam and we sandwich the mask between the wafer and a quartz window (blank chrome mask). The weight of the quartz window is sufficient to bring the flexible acetate mask into contact with the wafer. This was shown in Fig A2, but it was not described in the text. For the higher resolution features (using chrome mask), we have placed a 1 cm thick stainless steel ring to add weight. However, it is indeed possible to use a clamping rig in case sub 5 um features sizes are needed.

We have updated the manuscript (and Fig. 2) to reflect the above changes.

Regarding the 3D printing, we did not include SLA in the original text to avoid a lengthy introduction of the section, but accept that it should be mentioned as the alternative for 3D printer based low cost microfabrication. Aspects such as feature size, cost and waste are added to allow the reader to understand why we consider FFF as a more sustainable approach, at this point in time. We have updated the manuscript to explain our choice (lines 166-178).

We specifically use feature size rather than resolution, as variations in axial resolution are inevitable (particularly with negative features) depending on the 3D printer itself (stepper motor step angle, microstepping, belt/lead screw choice, pulley tooth count...). This said we understand that more detail should have been included to detail that positive features depend on nozzle diameter while negative features mostly depend on axis resolution. A footnote has been added to Table 1 to indicate this.

A description of the ironing process has been added (lines 196-201), while infill has been removed from Figure 3's caption as it had indeed been used correctly.

Regarding xurography,  as the reviewer points out, the improvement refers to thick PDMS (or other soft materials) layers. We have updated the manuscript (see lines 288-302) to emphasize the novelty of our process that consists in adding a tape on top of the soft layer to provide rigidity, prevent movement during cutting and enable easier post-cut transfer.

Reviewer 3 Report

I had a lot of trouble reviewing this paper. I always want to give constructive criticism where possible, especially if the paper goal is potentially useful. I do find the latter to be true, low cost microfab techniques can be useful, but in the end, my opinion is that the rigour is lacking for this to be accepted as a research paper. I think it would be better for the authors to focus on one technique at a time and do more serious and systematic studies on each, comparing the low-cost approach to standard (expensive) approaches so that a reasonable comparison of pros and cons can be obtained by the reader.

Author Response

We recognise that this paper is slightly unconventional and we apologise for the trouble it has created for the reviewer. However, we feel that the paper's goal is better served by the “toolbox” approach we present, as we feel it is more likely to appeal to people who do not have access to microfabrication tools, or have not considered it as an option. If we had limited the paper to a single technique (with more details), it would not show the true possibilities offered by microfabrication.

Moreover, as explained previoulsy, we would not be able to share the files/process/BOM/... had we decided to present a review paper. Therefore we feel that a research paper is the most appropriate way to disseminate our results.

Round 3

Reviewer 2 Report

I thank the authors for having addressed my concerns.